# Activation of the Rat P2X7 Receptor by Functionally Different ATP Activation Sites

**DOI:** 10.3390/cells14120855

**Published:** 2025-06-06

**Authors:** Fritz Markwardt, Malte Berthold, Sanaria Hawro Yakoob, Günther Schmalzing

**Affiliations:** 1Julius-Bernstein-Institute of Physiology, Martin-Luther-University, D-06097 Halle, Germany; malte.berthold@medizin.uni-halle.de; 2Institute of Clinical Pharmacology, RWTH Aachen University, D-52074 Aachen, Germany; sanaria_hawro@hotmail.de (S.H.Y.); gschmalzing@ukaachen.de (G.S.)

**Keywords:** purinoceptor, P2X7, activation, structure, kinetics, voltage clamp, model

## Abstract

The homotrimeric P2X7 receptor (P2X7R) contains three ATP^4−^ binding sites in its ectodomain. Here, we investigated the role of individual ATP^4−^ activation sites in rat P2X7R (rP2X7R) using trimeric concatemers consisting of either three wild-type subunits (7-7-7) or one to three subunits with ATP binding sites knocked out by the K64A mutation. Following expression in *Xenopus laevis* oocytes, ATP^4−^-elicited ion currents were recorded using the two-microelectrode voltage clamp technique. The 7-7-7 concatamer exhibited a biphasic ATP^4−^ concentration dependence, best fit by the sum of two Hill functions, confirming the existence of functionally distinct ATP^4−^ activation sites. The activation time course of the 7-7-7 was best approximated by the sum of a fast and a slow exponential saturating activation component. Similarly, deactivation exhibited both fast and slow exponential decay. Only one Hill function was required to best fit the ATP^4−^ concentration dependence of concatamers with only two or one ATP^4−^ binding sites, and their deactivation time courses largely lacked the slowly deactivating components. We conclude that the binding of one ATP^4−^ is sufficient for partial activation of the rP2X7R and that allosteric effects occur when all three ATP^4−^ binding sites are occupied, leading to distinct functional activation sites.

## 1. Introduction

The P2X7 receptor (P2X7R) is a non-selective cation channel predominantly expressed in cells of the immune and inflammatory systems. Like other P2X subunits, except P2X6 [1,2,3,4], P2X7 subunits efficiently assemble as homotrimers of three identical membrane-spanning subunits [5]. The P2X7R is activated by extracellular ATP, which is released from stressed or dying cells under pathophysiological conditions such as coagulation, inflammation and cell death. Extracellular ATP acts as a Danger-Associated Molecular Patterns (DAMPs) [6,7], signaling through the P2X7R to modulate immune responses.

High extracellular ATP concentrations are required for P2X7R activation for two reasons: (i) under physiological conditions, released ATP is largely complexed by millimolar extracellular Ca^2+^ and Mg^2+^, and (ii) the genuine P2X7R agonist, free-ATP (ATP^4−^), has a relatively low affinity, with an EC_50_ of ~100 µM in Ca^2+^- and Mg^2+^-free media [8,9]. In contrast, in Ca^2+^- and Mg^2+^-free solutions, the EC_50_ values of P2X1, P2X2, P2X3, and P2X4 receptors range only between 1.2 and 2.0 µM ATP, which corresponds to the fully ionized form ATP^4−^ as divalent cations were absent in these experiments [10]. For clarity and consistency, we explicitly use ATP^4−^ when referring to the free ATP concentration and ATP when referring to the total ATP concentration [11]. Since the extracellular solution in voltage clamp experiments was always divalent cation-free, in the Section 3, we consistently refer to the agonist ATP^4−^ as simply ATP.

The P2X7R is formed by three identical membrane-spanning subunits with large extracellular and intracellular domains [12]. The ectodomain contains three ATP binding sites, one at each subunit interface, which are located in the large extracellular domain at the interface between two adjacent subunits [4,12]. The binding pocket is formed by the head, upper body, and left flipper of one subunit and the lower body and dorsal fin of the adjacent subunit. Key residues involved in the coordination of ATP in a U-shaped configuration in the rat P2XR (rP2XR) include the positively charged residues K64, K66, and K311, as well as T189, which interacts with the adenine moiety, the β-phosphate group with N292, and R294 [4,12]. These residues are conserved across the P2X family [13]. K64 plays a crucial role in ATP binding for the P2X7 receptor by being centrally located among the three phosphate groups, forming hydrogen bonds with all of them, which is essential for ATP activation.

Although structurally identical, we have previously observed that the three ATP binding sites of the human P2X7R (hP2X7R) differ in their contributions to activation [14]. High- and low-affinity functional ATP activation sites were identified with dissociation constants of 10 and 300 µM ATP^4−^, respectively. The high-affinity activation site leads to an activation of approximately 10% of the maximum. For full activation of the hP2X7R, at least two ATP^4−^ molecules must bind with a K_D_ value of 300 µM ATP^4−^, as measured by single-channel recording [15].

To gain better insight into the putative sequential activation of rP2X7R through stepwise ATP binding at its three ATP activation sites, we generated genetically concatenated trimeric rP2X7R constructs with zero (7-7-7) to three (7^ko^-7^ko^-7^ko^) ATP binding site knockouts. Using long, flexible linkers, we observed negligible proteolytically cleaved side products from concatenated rP2X7 trimers but not from concatenated rP2X7 dimers. AlphaFold2 modeling revealed only minimal effects of the introduced linkers on the overall structure of the trimeric rP2X7 concatamers. Following expression in *X. laevis* oocytes, two-microelectrode voltage-clamp recordings on concatenated rP2X7 trimers revealed that the binding of a single ATP^4−^ molecule is sufficient to partially activate the P2X7R. When all three ATP^4−^ binding sites are occupied, allosteric effects result in distinct functional activation states.

## 2. Materials and Methods

### 2.1. Reagents

Standard chemicals and molecular biology reagents were obtained from Merck (Darmstadt, Germany) and New England Biolabs (Schwalbach, Germany), at analytical or higher grade, unless otherwise specified.

### 2.2. Cloning of rP2X7 Concatamers

The rP2X7R was first identified and cloned as a permeabilizing receptor [16]. Based on this sequence, we cloned the rP2X7R cDNA from total rat brain RNA using RT-PCR into the pNKS2 vector, as previously described [17,18]. rP2X7 concatamers were generated using a previously described strategy [19]. Briefly, NcoI and BspHI restriction sites were inserted at the 5′ and 3′ ends of the rP2X7R coding region, respectively, using the QuikChange site-directed mutagenesis protocol [20] with Phusion high-fidelity DNA polymerase and DpnI restriction endonuclease (both from New England BioLabs, Schwalbach, Germany).

To enable protein purification, the sequence encoding a double repeat of the Strep-tagII (WSHPQFEK), also known as Twin-Strep-tag or SIII tag or S3 [21], as abbreviated here, was inserted following codon 4 (alanine) of the rP2X7 to generate ^S3^rP2X7 (see Figure 1). The codon for asparagine (N) in the original Strep-tagII sequence (NWSHPQFEK) was omitted to avoid creating an N-glycosylation site that is utilized when luminally exposed [22]. The full S3 linker sequence (47 residues) introduced between copy 1 and 2 and copy 2 and 3 of the trimeric rP2X7 concatamer in single letter code is GGSGGGGSGGGGSGLMGSAWSHPQFEKGGGSGGGSGGSAWSHPQFEK. The residues are highlighted in the same colors as in the Protter cartoon in Figure 1. The leucine residue highlighted in yellow represents a cloning scare.

Additionally, to abolish ATP-dependent activation, a K^64^A mutation was introduced to generate ^S3^rP2X7^ko^. The crucial role of basic residues in ATP-dependent activation was first demonstrated for the rP2X1 receptor, where the K^68^A mutation resulted in a non-functional but plasma membrane-expressed channel with 1800-fold reduced ATP potency [23]. In the P2X7R, K64 is the residue homologous to K68 in rP2X1. The plasmid sequences of the rP2X4 and rP2X7 concatamer constructs are shown in the Appendix A.

All DNA constructs were designed and analyzed using Vector NTI Deluxe version 5, InforMax Inc. Introduced mutations were verified by comparing both the band patterns on agarose gels, generated by digestion with selected restriction enzymes, and commercial DNA sequencing results (Eurofins Genomics, Ebersberg, Germany) to the in silico predictions made using Vector NTI predictions. The exact same modular assembly strategy was used to clone the concatenated ^S3^rP2X4 homotrimer.

Full-length coding sequences, including appropriate flanking sequences, were excised using HindIII and BspHI and ligated in-frame between the HindIII and NcoI sites of the parental rP2X7 plasmids to generate concatenated ^S3^rP2X7 and ^S3^rP2X7^ko^ homo- and heteromultimers. Note that NcoI and BspHI sites can be ligated together, but the resulting hybrid cannot be cleaved at the ligated sites by either enzyme. This principle enables a modular assembly approach to generate ^S3^rP2X7- and ^S3^rP2X7^ko^-containing concatamers in any desired order.

cRNA was synthesized as previously described in detail [24] including co-transcriptional incorporation of the anti-reverse cap analog (ARCA Cap Analog, m_2_^7,3′−O^Gp_3_G; NU-855; Jena Bioscience, Jena, Germany) to ensure the correct orientation at the ATG start codon of the cRNA [25]. Polyadenylation, which enhances cRNA translation in *X. laevis* oocytes [26], was co-transcriptionally encoded by the pNKS2 vector used [17]. The quality of the cRNA was assessed by ethidium bromide-stained agarose gel electrophoresis and the spectrophotometric determination of concentration and the 260/280 nm absorbance ratio, which ranged from 2.00 to 2.04, indicating minimal protein contamination.

### 2.3. Obtaining X. laevis Oocytes for In Vitro Experiments

Female *Xenopus laevis* were purchased from certified breeders. The animals were housed and ovariectomized under tricaine immersion bath anesthesia, following protocols approved by the local animal welfare committees in Halle (Germany, reference no. Az. 203.42502-2-1493 MLU) and Düsseldorf (Germany, reference no. 8.87-51.05.20.10.131) for experiments performed in Halle and Aachen, respectively. All animal procedures were conducted in compliance with EC Directive 86/609/EEC and are reported in accordance with the ARRIVE guidelines, as far as applicable to our exclusively in vitro experimental design (see Appendix A).

### 2.4. Electrophysiological Characterization of rP2X7 Concatamers by Two-Electrode Voltage Clamp

Freshly isolated *X. laevis* oocytes were defolliculated with collagenase. Stages V–VI oocytes (according to Dumont) were selected and injected with cRNAs encoding the indicated rP2X7 constructs; the oocytes were then cultured for 2–3 days at 21 °C [18]. To ensure comparable amplitudes of ionic currents, the cRNA of the rP2X7 concatamer 7-7-7 (without knockout) was diluted 1:10 to a final concentration of approximately 0.1 µg/µL in voltage clamp experiments.

### 2.5. Biochemical Visualization of the Intactness of Plasma Membrane-Bound P2X7 Concatamers

S3-affinity-tagged rP2X7R subunits were expressed in *X. laevis* oocytes by cRNA injection as monomers, concatenated dimers, or concatenated trimers. Typically, 15–20 oocytes were injected per experimental group. On day 3 post-injection, the cell surface of intact oocytes was covalently labeled with membrane-impermeable fluorescent IRDye® 800CW NHS-Ester (LI-COR®, Biosciences, Lincoln, NE, USA) as previously described [27]. Morphologically healthy oocytes, 10 per group, were then selected for purification using stereomicroscopy. A digitonin extract (1% digitonin) of the oocytes was prepared [26], from which the indicated proteins were purified by Strep-Tactin chromatography [5]. The purified proteins were then denatured by incubation with 0.2% SDS and 10 mM DTT at 37 °C in a Coomassie-free sample buffer and resolved on a clear SDS-PAGE gradient gel. Coomassie was avoided because it would strongly suppress fluorescence signals. Instead, fluorescence scanning of the wet SDS-PAGE gel using a Bio-Rad ChemiDoc MP visualized the 800CW-stained plasma membrane-bound proteins in the near-infrared channel and the covalently blue-stained Precision Plus Protein Standards (Bio-Rad, Hercules, CA, USA) in the red channel. The intensity of the 800CW-stained bands was quantified using BioRad Image Lab 6.0.1.

### 2.6. Two-Electrode Voltage-Clamp Recording (TEVC)

The voltage clamp protocol was as described previously [18]. In brief, microelectrodes filled with 3 M KCl, with resistances of 0.8–1.6 MΩ, were impaled into oocytes superfused with oocyte Ringer’s solution (ORi: 100 mM NaCl, 2.5 mM KCl, 1 mM CaCl_2_, 1 mM MgCl_2_, 5 mM HEPES, pH 7.4). Currents were recorded at room temperature using an OC-725C oocyte clamp amplifier (Warner Instruments, Hamden, CT, USA). The signals were filtered at 100 Hz and sampled at 85 Hz, with a holding potential of −40 mV. Switching between the different bathing solutions was achieved within <1 s by a set of computer-controlled magnetic valves using a modified U-tube technique [24].

Measurements of the rP2X7R-dependent currents were performed in bathing solutions consisting of 100 mM NaCl, 2.5 mM KCl, and 5 mM HEPES, pH 7.4. This Ca^2+^-free solution was supplemented with 0.1 mM flufenamic acid to block the conductance evoked by external divalent cation removal. To test for rP2X7R-dependent changes in the cell membrane conductance, the Ca^2+^-free solution was replaced by the same solution with added ATP.

The data were stored and analyzed on a personal computer. For ion current recording and analysis, a custom software system developed in our department was used (Superpatch 2000, SP-Analyzer by T. Böhm). The SigmaPlot program (version 12.5, SPSS) was used for non-linear approximations and graphical representations of the data. Statistical data were expressed as mean ± SEM and analyzed via one-way ANOVA. The statistical significance of the differences between means was tested using multiple *t*-tests with Bonferroni correction, performed using the SigmaPlot program. Significance was set at *p* < 0.05.

### 2.7. Alphafold2 Structure Predictions of Concatenated and Non-Concatenated rP2X7 Homotrimers

To potentially detect gross structure disturbances caused by the introduced 126-residue intersubunit linker and Strep-tag sequences, the full amino acid sequence of the 1911-residue rP2X7 7-7-7 concatamer was run on AlphaFold2 ColabFold v1.5.5 with the following settings: Template mode: none; MSA mode: mmseq2_unifre_env; Pair mode: unpaired–paired; Advanced settings: Model type: alphafold2_multimer_v3; Number of recycles: 6; Recycle early stop tolerance: 0.5; Relax max iterations: 200; Pairing strategy: greedy [28]. The models obtained were then visualized via their PDB files using PyMOL version 2.6.2 (Schrödinger, LLC, New York, NY, USA). For a direct comparison, we used the same settings to generate an AlphaFold2 model of the non-concatenated rP2X7 protein, despite the availability of cryo-EM structures [4,12].

### 2.8. Positioning of the Alphafold2 Models of the rP2X7 and 7-7-7 Concatamer in the Membrane

The membrane boundaries of the Alphafold2 modeled structures of the rP2X7 7-7-7 concatamer (total length 1911 residues) and the wild-type non-concatenated rP2X7 homotrimer (total length 1785 residues) were determined by uploading the PDB files to the PPM2.0 Web Server (https://opm.phar.umich.edu/ppm_server2_cgopm, accessed on 05 June 2025), which is part of the OPM (Orientations of Proteins in Membranes) server (https://opm.phar.umich.edu/, accessed on 05 June 2025), using the following conditions: “Membrane type: Mammalian plasma membrane”, “Allow curvature” set to “Yes”, and topology with the “N-terminus in” [29]. The resulting PDB files, which included the membrane boundaries, were then downloaded and visualized using PyMOL 3 (Schrödinger LLC).

### 2.9. Comparison of the Alphafold2-Modeled Structures of the rP2X7 and 7-7-7 with PyMOL 3

To compare the structures of the two AlphaFold2-modeled proteins, we superimposed their structures in PyMOL 3 using the align command to visualize and quantify their spatial alignment. The root mean square deviation (RMSD) was calculated to determine the average distance between corresponding atoms in the two structures, serving as a measure of their similarity. The structures were visualized in a cartoon to illustrate the secondary structure and spatial arrangement of the proteins and color-coded to highlight differences in conformation or the position of amino acids.

## 3. Results

### 3.1. Evaluation of the Proteoloytic Stability of Homodimeric and Homotrimeric rP2X7 Concatamers

The N- and C-terminal ends of P2X receptors are generally located intracellularly, allowing for the expression of covalently linked subunit concatamers through genetic N- to C-terminal linkage. Based on our previous observation that rP2X1 concatamers can be proteolytically unstable when expressed in *X. laevis* oocytes [19], we genetically linked the rP2X7 subunits N- to C-terminally with long GGS-linkers to improve concatamer stability by increasing flexibility. We also included intervening “Strep-double tags” to enable their purification.

Figure 1 illustrates the transmembrane folding of the wild-type (wt) homotrimeric ^S3^rP2X7 and ^S3^rP2X4 concatamers, generated using the online tool Protter [30], available at (https://wlab.ethz.ch/protter, accessed on 16 May 2025). The residues equivalent to the K64 residue, highlighted in blue in the ectodomain of each rP2X7 subunit, are completely conserved across the P2X family. Mutation of this lysine residue to alanine abolishes ATP-dependent activation, as first demonstrated for the corresponding rP2X1R^K68A^ mutant [23].

Figure 2A shows the full-length AlphaFold2-modeled structure of the 7-7-7 concatenated homotrimer, comprising 585 residues per monomer, connected by GGS linkers and Strep tags highlighted in the cytoplasmic domain (for the exact sequences, see Figure 1A). Figure 2B depicts a non-concatenated rP2X7R AlphaFold2-modeled homotrimer, consisting of the same 585 genuine rP2X7R residues per monomer but lacking the cytoplasmic GGS linkers and Strep tags present in the concatamer in Figure 2A.

To visualize the potential impact of the engineered linker and Strep-tag sequences on the overall structure, we superimposed both structures using PyMOL 3 (Figure 2C). As a measure of their similarity, we calculated the root mean square deviation (RMSD), which quantifies the average distance between corresponding atoms in the two structures [31]. The very low RMSD value of 0.481 Å between the two structures indicates a very high degree of structural similarity, suggesting that the linkers and Strep tags do not significantly alter the overall structure of the concatamer. Although the linkers may still influence protein stability and flexibility, potentially affecting biological activity, the very low RMSD supports the conclusion that the overall structure of the concatamer remains virtually identical to the wild-type receptor.

**Figure 2 cells-14-00855-f002:**
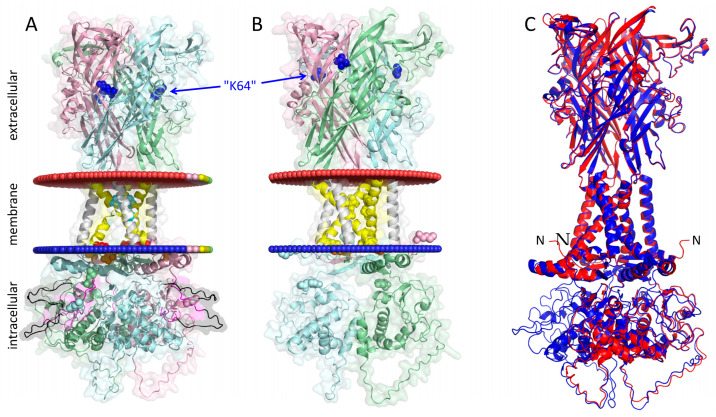
AlphaFold2 predicted structures of assembled and concatenated rP2X7 constructs. The homotrimeric rP2X7R concatamer (7-7-7) (**A**) and the non-concatenated rP2X7R (**B**) were generated using AlphaFold2. Modeling details are provided in the Section 2. The membrane boundaries were determined using the “OPM” (Orientations of Proteins in Membranes) server (https://opm.phar.umich.edu/, 5 June 2025) with the following conditions: “Membrane type: Mammalian plasma membrane”, “Allow curvature: Yes”, and “Topology: N-terminus in” [29,32]. The three concatenated subunits are depicted in pale green, pale cyan, and light pink, except for TM1 and TM2, which are shown in light gray and yellow, respectively, to facilitate identification. The color code is largely consistent with Figure 1; the cation-selectivity-determining residues D14 and D352 are colored red, the gating residues S339 and S342 are shown in cyan, and the ATP-binding residue K64 (knocked out in this work by mutation to alanine, corresponding to K96 in concatamer numbering) is colored blue. The S3 tag sequences (WSHPQFEK) are colored magenta, and the GGS linker sequences are shown in black, differing from Figure 1, where they are gray. These features are visible as lines. The membrane boundaries were determined by uploading the PDB file to the PPM2.0 Web Server (https://opm.phar.umich.edu/ppm_server2_cgopm, accessed on 5 June 2025) and visualizing the resulting downloaded PDB using PyMOL (Schrödinger LLC). (**C**) Structural comparison of the AlphaFold2 modeled 7^wt^-7^wt^-7^wt^ rP2X7 concatamer and the non-concatenated rP2X7R using the align command of PyMOL 3 yielded a root mean square deviation (RMSD) value of 0.481 Å. This suggests that concatamerization does not significantly alter the overall structure.

In addition, to verify their biochemical integrity, we expressed homodimeric and homotrimeric rP2X7 concatamers by cRNA injection in *X. laevis* oocytes for three days, purified them by StrepTactin chromatography, and resolved them by SDS-PAGE. The concatameric constructs, including the single, double, and triple (7^ko^-7^ko^-7^ko^) knockouts^o^, were detected at the oocyte´s plasma membrane three days after injection of the corresponding cRNAs, albeit at a significantly lower level (Figure 3, lanes 3–24) compared to the parental monomers (Figure 3, lanes 1–2). The expression level was not significantly influenced by the K64A knock-out mutation.

The positive result is that the concatenated rP2X7 homotrimers migrated virtually completely as homotrimers (Figure 3, lanes 11–20 and 23–24). This view was further substantiated by quantitative scanning of the IR800 dye covalently bound to the rP2X7 concatamers expressed in the plasma membrane, which revealed little to no cleavage products migrating as a dimer or monomer (Appendix A).

However, significant amounts of monomeric cleavage products were detected when homodimeric concatamers were expressed (Figure 3, lanes 3–8). The expression of homodimers resulted in obvious proteolytic cleavage of a fraction of the expressed homodimers, as evident from the significant amounts of apparently rP2X7 monomers that were absent when concatenated homotrimers were expressed (Figure 3, lanes 9–20 and 23–24; see also the quantitative evaluation of Figure 3 in Appendix A). A likely explanation is that the monomers are part of a frustrated assembly as a pseudo-tetramer, consisting of two dimeric rP2X7 concatamers, which results in ER-associated degradation [33] of the subunit not integrated into the trimeric structure, akin to a “leave-one-out” scenario. The final “heterotrimeric” construct (labeled in Figure 3 as 1–2) is exported to the plasma membrane, where it is labeled by the membrane-impermeable IR800 dye. Most likely, the visible concatenated homodimers and the monomer (marked with an asterisk) also existed as pseudo-homotrimers before they were purified and partially denatured by SDS. Taken together, misassembly due to a non-native structure (here a tetramer formed of two dimers), combined with ER-associated degradation can lead to a significant misinterpretation of functional experiments when not combined with appropriate biochemical controls.

### 3.2. Electrophysiological Characterization of Knockouts of Individual ATP Binding Sites in Trimeric rP2X7 Concatamers

The localization of the K64A mutation, which knocks out ATP binding sites in the rP2X7 ectodomain, is schematically illustrated in Figure 4A. The drastic reduction in ATP-induced currents for the triple knockout, by a factor of approximately 0.03 across all tested ATP concentrations (Figure 4B), indicates a significant decrease in the ATP affinity of the mutated agonist binding sites.

The similar physical expression of all concatemeric constructs (Figure 3) is reflected by comparable ATP-induced currents when the cRNA of the 7^wt^-7^wt^-7^wt^ concatamer was diluted 1:10 for injection into *X. laevis* oocytes (Figure 5D). This suggests that knockout mutations do not drastically affect channel function. Further studies using single-channel recordings would be required to resolve potential effects on open probability or conductance.

Typical time courses of ATP-induced rP2X7 concatamer-mediated currents are shown in Figure 5A–C. The best fit was always achieved by a model with the sum of two exponentially saturating components for activation and two exponentially decreasing current components for deactivation of the current (Equations (1) and (2), see legend).

The main difference between the “wild-type” concatamer 7^wt^-7^wt^-7^wt^ and the shown knockout constructs is the presence of a large, slowly deactivating current component in addition to the fast component (Figure 5A). This suggests the dissociation of bound ATP from binding sites with high and low affinity, respectively, indicating the existence of functionally different ATP activation sites. However, the slowly deactivating component is strongly reduced in all knockout concatamers (Figure 5E). The detailed fitted parameters are shown in Figure 6.

To gain further insight into the function of the ATP activation sites, we constructed concentration–response curves for the activation of ATP-dependent, whole-cell ion currents for all concatemeric constructs. We then attempted to approximate these curves using different Hill plots based on Equations (4) and (5). The models resulting from these equations provided the best fits, yielding the highest correlation coefficients and lowest sums of squares of errors [34].

The curves for the concatamers consisting of either ^S3^rP2X7 subunits (Figure 7A) or ^S3^rP2X4 subunits (Figure 7D) display biphasic behavior, again indicating functionally different ATP activation sites. Approximation of these data using a single Hill plot resulted in Hill coefficients significantly lower than unity, indicating negative cooperativity between the ATP binding sites for these concatamers (Figure 7A,F). Only the sum of two Hill functions resulted in the best fit for the 7-7-7 concatamer, yielding functional apparent K_D_ values of about 10 and 1800 µM. In contrast, the concentration–response curves for the concatamers with one (see, for example, Figure 7B) or two (Figure 7C) ATP binding site knockouts were best fit with single Hill curves. The K_D_ values of the concentration–response curves for the single knockout constructs were statistically equivalent to the value for the high-affinity ATP binding site of the 7-7-7 concatamer.

Although the single knockout concatamers contain two intact ATP binding sites, the approximation by a Hill function with a free Hill coefficient (n) yielded values not significantly different from 1 (Figure 7F). This indicates the absence of cooperativity between the remaining two activation sites. The concentration dependency of the inward current amplitudes of the double knockout concatamers was also best fit by a single Hill curve, as expected for constructs with only one functional ATP binding site remaining. The corresponding K_D_ values were equal to the K_D_ value of the low-affinity ATP binding site of the 7-7-7 concatamer, with the exception of the 7^ko^-7-7^ko^ concatamer, where fitting with a Hill function with a free Hill coefficient (Equation (5)) resulted in a Hill coefficient significantly larger than 1 (Figure 7E).

## 4. Discussion

### 4.1. Homotrimeric rP2X7 Concatamers, in Contrast to Their Homodimeric Precursors, Are Proteolytically Stable

We have previously observed that P2X1 concatamers not only reduce overall stability but also result in the appearance of small byproducts that we interpreted as unwanted cleavage products [19]. Significant proteolytic cleavage is not generally observed with P2X concatamers, as concatenated P2X2 receptor subunits resulted in only negligible, if any, formation of dimers [35]. To remove possible constraints by linking the ends of P2X7 subunits together, we inserted very long and flexible linkers combined with multiple Strep Affinity tags (see Figure 1) that allowed us to purify not only intact concatamers but also shorter possible cleavage products.

### 4.2. Binding of One ATP Is Sufficient to Open the rP2X7 Channel Pore

The number of ATP molecules required to open a single P2X receptor channel remains a subject of intense research and discussion [15,35,36,37,38,39,40,41,42,43]. Structure models of P2X receptors show apo-closed and fully open states for P2X4R [44], P2X3R [45], and P2X7 [12], including calculated simulations of symmetrical pore opening. This aligns with P2X kinetic models deduced from voltage clamp measurements [36,39,40].

However, our findings and those of Gusic et al. [43] indicate that binding of one ATP molecule is sufficient to open the rP2X7 ion channel, suggesting an asymmetric pore opening mechanism. Analysis of the P2X receptor activation time course and its ATP concentration dependency has led to the conclusion that the binding of just one or two ATP molecules can induce P2X receptor-dependent ion currents [35,37,38,42]. Our previous detailed investigation of the kinetics of single hP2X7 receptor channels resulted in a model where binding of two ATP molecules is necessary and sufficient to fully open the ion channel [15]. Given the existence of three identical ATP activation sites, it can therefore be hypothesized that the binding of the first ATP molecule either does not lead to any channel opening or generates a very small current due to incomplete channel pore formation, which may have escaped detection in single-channel recordings.

The structure of the cytoplasmic ballast of the rP2X7R, which contains the N- and C-terminal ends of the subunits [12], might be altered by concatamerization due to the covalent connection of N- and C-termini from different subunits, thereby potentially altering the channel kinetics. However, the typical biexponential activation and deactivation time courses of the wt P2X7R channels are preserved in the concatamers (see Figure 3 and Figure 4). Furthermore, structural models of the closed and open hP2X7R ion channel indicate that the structure of the cytoplasmic ballast, including its intracellular side windows crucial for cation selectivity, remains largely unchanged between open and closed channel pores [46]. This is further confirmed by structural alignments of our AlphaFold2 3D models of the wt and concatenated rP2X7R, which reveal only minor conformational changes due to concatamerization, as verified by the low RMSD value of 0.481 Å (see Figure 2). An RMSD value below 1 Å indicates very high similarity between two structures, supporting the conclusion that concatamerization had only minor effects on the overall structure [47]. In contrast, the combination of concatamerization and the use of K^64^A knockout subunits affects the activation kinetics.

Unlike the experiments reported by the Benndorf group [43], some of our knockout constructs exhibit slow desensitization characterized by a positive I_act,slow,6s,rel_ current, rather than the slowly exponentially growing inward current observed for the wild-type rP2X7R and the 7^wt^-7^wt^-7^wt^ concatamer (see Figure 6C). Since the activation of P2X7Rs measured in outside–out patches at saturating ATP concentrations occurs within 20 ms (rP2X7R [43]) or 10 ms (hP2X7R [15]), the activation measured in our whole-cell experiments is strongly influenced by the slower solution change compared to patch measurements. The cause of the slow P2X7R-dependent current activation in whole-cell current measurements remains unclear [6] and deserves further investigation.

### 4.3. Functionally Distinct Activation Sites at P2X Receptors

A symmetrical opening of P2XR channels after binding of all three ATP molecules to the structurally equal binding sites would result in a dependence of the ion current on ATP concentration that could be described by a single Hill function. However, this was not the case for the concentration dependency of concatemerized rat P2X7 and P2X4 receptors, as shown here. Instead, approximation by a single Hill function resulted in low correlations and Hill coefficients smaller than 1 (Figure 7A,D,F). The assumption of functionally different activation sites is supported by the clearly biphasic deactivation of the ATP-dependent currents observed for the 7^wt^-7^wt^-7^wt^ concatamer, which is absent for the rP2X7 concatamers with ATP binding site knockouts (Figure 5).

Biphasic ATP concentration dependencies, which required approximation by the sum of two Hill functions, have already been reported for human P2X7 receptors [14,48] and mouse P2X7 receptors [49,50]. An explanation for the differing affinities of the three structurally equal P2X7R activation sites is the assumption of negative cooperativity between these sites. This would imply that the binding of the first and possibly also the second ATP molecule has an allosteric effect on the third binding site, thereby reducing its binding affinity, as also shown here (Figure 7). In accordance with this, the concentration dependencies of the rP2X7R concatamers with two intact ATP activation sites were approximated by single Hill functions with K_D_ values indistinguishable from the high-affinity K_D_ value of the 7-7-7 concatamer (Figure 7E).

As a logical conclusion from this argument, one would expect that concatamers with only one intact ATP binding site would become activated at low ATP concentrations too, i.e., low K_D_ values. However, this is not the case (Figure 7). Instead, these concatamers have K_D_ values similar to the low-affinity K_D_ of the 7-7-7 concatamer. The cause of this remains unclear. Either certain allosteric effects of the two mutated ATP binding sites lead to a low affinity of the remaining intact activation site, or strong positive cooperativity of two ATP bindings at the unmodified P2X7R increases the functional ATP affinity. However, the approximated Hill coefficients near 1 for the single mutation concatamers (7^ko^-7-7, 7-7^ko^-7, and 7-7-7^ko^) argue against strong cooperativity.

On the other hand, the Hill coefficient alone appears to be a weak argument for making a statement about cooperativity [51]. For example, the Hill coefficient greater than 1 for the 7^ko^-7^wt^-7^ko^ concatamer, which has only one intact activation site, is similar to what was found for a 2^ko^-2^ko^-2^wt^ P2X2 concatamer [35]. Recent reports have demonstrated that, depending on experimental conditions, Hill coefficients near 1 are observed for all P2X7 concatamers [43]. For concatemerized P2X2 receptors, Hill coefficients range between 1 and 2 [40,52], are greater than 2 [36,53,54], or indicate negative cooperativity for ATP binding but positive cooperativity for the subsequent structural change [42]. This variability highlights the uncertainty of deriving the number of ATP bonds required for channel activation from the fitted Hill coefficients.

Negative cooperativity between ATP binding sites, associated with functionally distinct activation sites, is presumably a property of P2X receptors other than the P2X7R too, as exemplified by the need for two Hill functions to fit the ATP concentration dependency also of rP2X4 receptors, as shown here (Figure 7D). This phenomenon may have been overlooked in studies that did not investigate concentration dependencies up to 10 mM ATP^4−^, especially for P2X receptors with apparently higher ATP^4−^ affinity than the P2X7 receptor [42,52,55,56,57]. Another possibility is that either the high or the low ATP affinity component may be too small to become evident in voltage clamp experiments.

## Figures and Tables

**Figure 1 cells-14-00855-f001:**
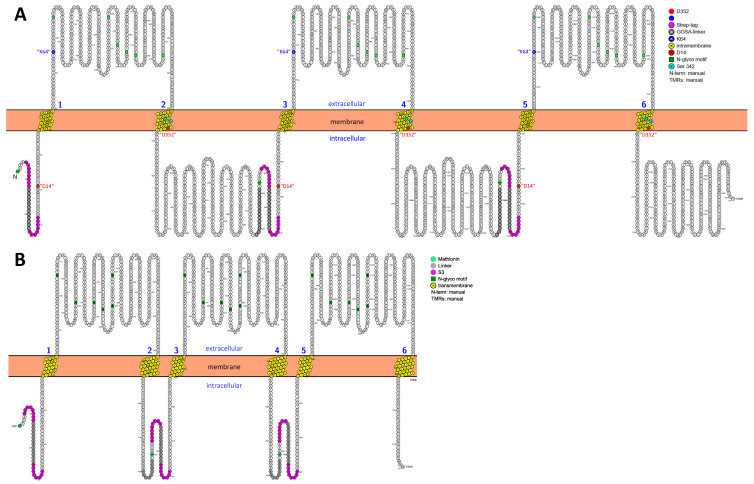
Transmembrane topology of homotrimeric ^S3^rP2X7 and ^S3^rP2X4 concatamers. Membrane-embedded residues are shown in yellow, N-glycosylation sites are shown in green, the S3 tag sequences (WSHPQFEK) are colored magenta, and the long and flexible GGS linker sequences are shown in gray. (**A**) rP2X7: The cation-selectivity-determining residues D14 and D352 are shown in red (D41 and D379 in concatamer numbering), the gating residues S339 and S342 are shown in cyan (S366 and S369 in concatamer numbering), and the ATP-binding residue K64 (knocked out by mutation to alanine singly, doubly, or up to triply in the 7^ko^-7^ko^-7^ko^ concatamer) is shown in blue (K96 in concatamer numbering). Crucial residues in the context of this work are labeled using a hyphenated notation (e.g., “K64”), where the number refers to the established position in the native rP2X7 sequence in order to distinguish between the positions from those in the concatamer. (**B**) ^S3^rP2X4: residues homologous to those in rP2X7 in rP2X4 are D16, D354 (D47 and D385 in concatamer numbering), and the putative gating residues S341 and A344 in cyan (S372 and S375 in concatamer numbering). The figures were generated using the open-source tool Protter (https://wlab.ethz.ch/protter/start/, accessed on 16 May 2025).

**Figure 3 cells-14-00855-f003:**
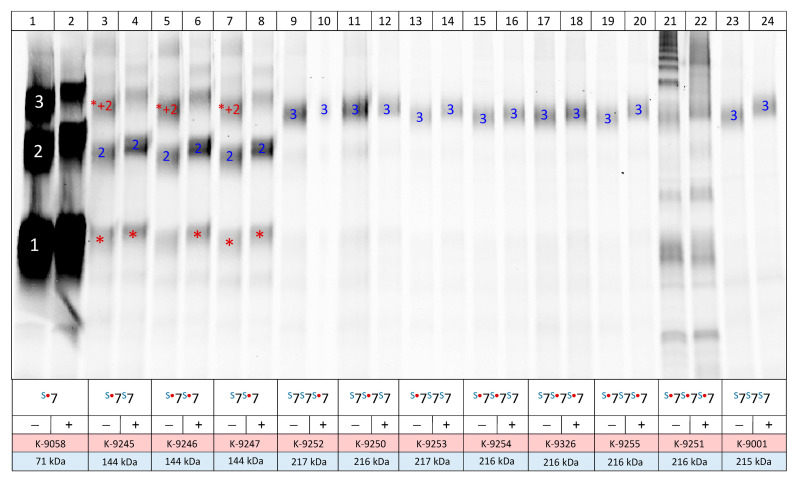
SDS-PAGE analysis of the affinity-purified rP2X7 concatamers to control their full-length integrity. Oocytes injected with the indicated cRNAs were incubated for 3 days at 21 °C and then surface-labeled with the amine-reactive dye IR800 NHS ester (LI-COR). The oocytes were extracted with digitonin, purified through their Strep-tag(s) using Strep-Tactin Sepharose (IBA Lifesciences), and resolved by SDS-PAGE. The SDS-PAGE gel was scanned wet for IR800 fluorescence using a LI-COR scanner at 800 nm. The proteins were visualized and quantified using Image Lab 6.01 (Bio-Rad Laboratories). Lanes 1 and 2, which show the parental ^S3^rP2X7, appear overexposed compared to all the other lanes because identical Image Lab 6.01 settings were used. This approach was used to demonstrate the differences in expression levels between the non-concatenated and concatenated homodimeric and homotrimeric constructs. Lanes 21 and 22, which cannot be evaluated due to the presence of leaky oocytes, are also shown to maintain the integrity of the scan. In the legend, “S” signifies the Strep-tag, the red dot signifies the K^64^A mutation of the respective subunit, and “7” represents the rP2X7 monomer. Red stars indicate monomeric cleavage products. The symbols “1”, “2”, and “3” denote monomeric, dimeric, and trimeric constructs, respectively. The notation “*+2” indicates that the band represents a non-covalent trimeric assembly consisting of an expressed homodimer (2) plus a monomer (*), which is a monomeric cleavage product of the concatenated homodimer. The incubation with 20 mM DTT is indicated by “+”. Quantitative scans of the fluorescent bands are shown in Appendix A.

**Figure 4 cells-14-00855-f004:**
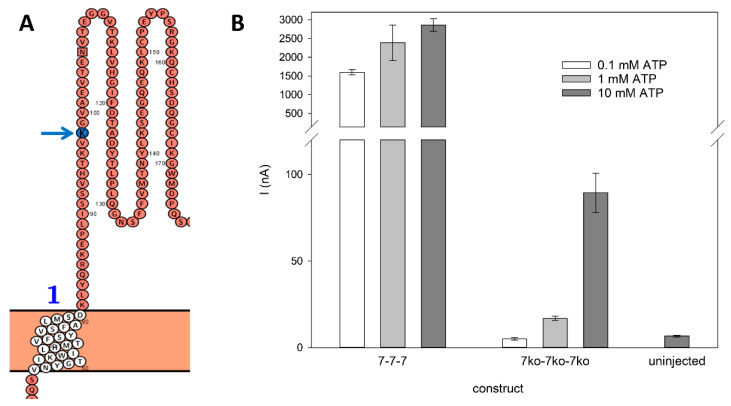
Effect of ATP binding site knockout on rP2X7 activation. (**A**) Enlargement of a region of the extracellular domain of the first rP2X7 subunit in the homotrimeric rP2X7 concatamer. The arrow highlights the location of the K64 residue in the wild-type rP2X7 monomers. Mutation of this residue to alanine creates the ATP binding site knockout (7^ko^). Note: numbering in this panel corresponds to the modified construct, which includes linker and Strep tag insertions. “1” denotes the first transmembrane domain according to Figure 1. (**B**) Functional impact of the triple K64 knockout (all three subunits mutated) on currents induced by the indicated concentrations of ATP, compared to the wild-type homotrimeric rP2X7 concatamer. ATP concentrations tested are indicated. The position of K64 in the 3D structure is shown in Figure 2.

**Figure 5 cells-14-00855-f005:**
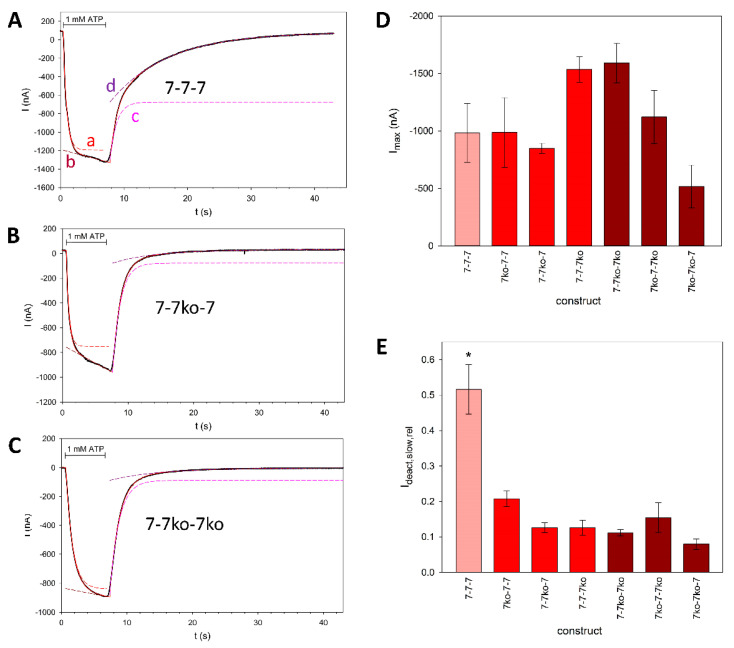
Kinetics of rP2X7(^ko^) constructs. (**A**–**C**) Typical current traces of the rP2X7 concatamers 7-7-7 (**A**), 7-7^ko^-7 (**B**), and 7-7^ko^-7^ko^ (**C**). The time of ATP application is indicated. The non-linear fitted model (red solid line) was used for (a) ATP-dependent activation: $I={I_{0}+A}_{1act}\cdot \left(1-e^{-R_{1act}\cdot t}\right)+A_{2act}\cdot \left(1-e^{-R_{2act}\cdot t}\right)$, (1), where I_0_ is the steady-state current before ATP application, A_1act_ and A_1act_ are the amplitudes, and R_1act_ and R_1act_ are the rate constants of the fast and slow activating current components, respectively, and (b) deactivation after ATP washout: $I=A_{1deact}\cdot e^{-R_{1deact}\cdot t}+A_{2deact}\cdot e^{-R_{2deact}\cdot t}+I_{\infty }$, (2), where A_1deact_ and A_2deact_ are the amplitudes, R_1deact_ and R_2deact_ are the rate constant of the fast and slow deactivating current components, respectively, and I_∞_ is a non-deactivating current component. The components of fast and slow activation (“a” and “b”) and deactivation (“c” and “d”) are shown as dashed lines. The corresponding data are given in Table 1. (**D**) Inward currents evoked by application of 1 mM ATP. The amplitudes are not significantly different. (**E**) Relative amplitude of the slowly deactivating current, as calculated by I_deact,slow,rel_ = A_2deact_/(A_1deact_ + A_2deact_) (3). As indicated by an asterisk, I_deact,slow,rel_ for the 7-7-7 construct is significantly different from the others. Measurements were performed on N = 6 to 10 oocytes.

**Figure 6 cells-14-00855-f006:**
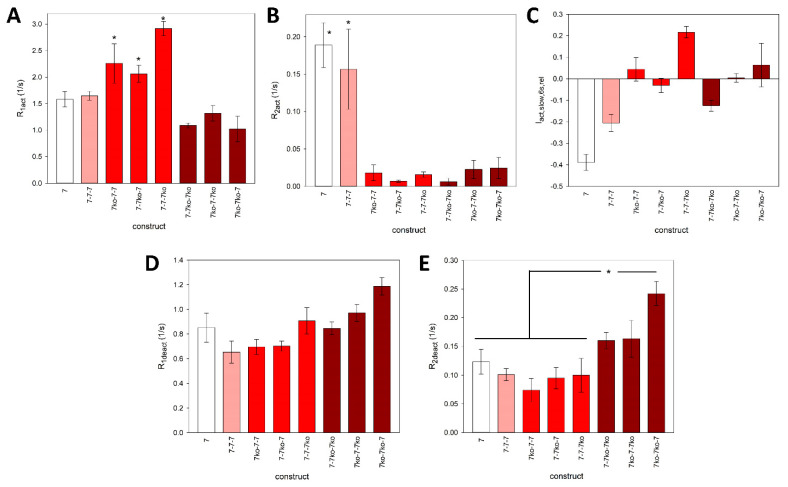
Parameters of the fits to the rP2X7 concatamer kinetics. The approximated high- and low-rate constants for activation (**A**,**B**) and deactivation (**D**,**E**) and the relative contribution of the slow exponentially activating component within 6 s of 1 mM ATP application (**C**) are shown. Means marked with an asterisk are significantly different from the unmarked means. Measurements were obtained from N = 4 to 20 oocytes.

**Figure 7 cells-14-00855-f007:**
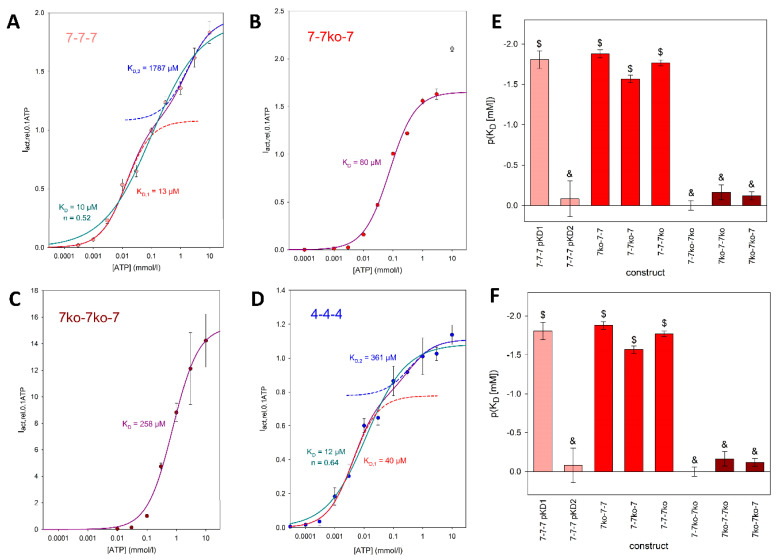
Agonist concentration dependence of the maximal amplitudes of ATP-induced currents mediated by trimeric rP2X7 concatamers. Examples of concatamers with (**A**) 0 (7-7-7), (**B**) 1 (7-7^ko^-7), and (**C**) 2 (7-7^ko^-7^ko^) knockouts of the ATP binding sites are shown in addition to (**D**) the 4-4-4 concatamer. The relative currents were best approximated by the sum of two functions (with Hill coefficients 1 and 2) for (**A**,**D**), $I_{rel}(\left[ATP\right])=\frac{I_{\;[ATP]}}{I_{0.1\;ATP\;before}}=\frac{I_{max,1}}{1+\frac{K_{D,1}}{\left[ATP\right]}}+\frac{I_{max,2}}{1+\frac{K_{D,2}}{\left[ATP\right]}}$ (4), and by one Hill function for (**B**,**C**). Here, I_[ATP]_ is the maximal inward current during ATP application, *I*_0.1 ATP before_ is the maximal inward current during the application of 0.1 mM ATP 2 min before, K_D_ values are the apparent ATP dissociation constants of the ATP binding sites, and *I*_max_ values are the maximal relative currents. The outlier at 10 mM ATP^4−^ for the 7-7^ko^-7 concatamer (**B**) was discarded. At these high concentrations, the large *I*_rel_ is presumably generated by the binding of ATP to the knockout activation site, leading to increased activation of the construct. (**E**) pK_D_ values of the investigated rP2X7 concatamers. Mean values with equal symbols are statistically not different. (**F**) Hill coefficients (n) of single Hill plots of the ATP concentration dependence of the concatamers. The concentration dependencies of *I*_rel_ were approximated by $I_{rel}(\left[ATP\right])=\frac{I_{\;[ATP]}}{I_{0.1\;ATP\;before}}=\frac{I_{max}}{{\left(1+\frac{K_{D,1}}{\left[ATP\right]}\right)}^{n}}$ (5). In (**A**,**D**), the fits using Equation (5) and the corresponding K_D_ values are also shown. Mean values of the concatamers 7-7-7 and 7^ko^-7-7^ko^ are significantly different from the others. Measurements were obtained from N = 5 to 10 oocytes.

**Table 1 cells-14-00855-t001:** Kinetic parameters of the ionic currents shown in Figure 5A–C.

Concatamer	A_1act_ (nA)	R_1act_ (1/s)	A_2act_ (nA)	R_2act_ (1/s)	A_1deact_ (nA)	R_1deact_ (1/s)	A_2deact_ (nA)	R_2deact_ (1/s)
7^wt^-7^wt^-7^wt^	−1286.7	1.81	−6572.4	0.00284	−626.6	0.95	−756.1	0.11
7^wt^ -7^ko^-7^wt^	−783.1	2.3	−7595.2	0.00382	−872.5	0.84	−108.7	0.15
7^ko^-7^ko^-7^wt^	−838.2	0.98	−3485.1	0.00259	−785.1	0.72	−86.1	0.13

## Data Availability

The raw data supporting the conclusions of this article will be made available by the authors on request.

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
