# Peer review of "Activation of the Rat P2X7 Receptor by Functionally Different ATP Activation Sites"

_cells, 2025, doi:10.3390/cells14120855_

Round 1
Reviewer 1 Report
Comments and Suggestions for Authors
The purpose of this study was to understand how different binding sites for ATP affect P2X7 activation. The study utilizes three concatenated trimeric rP2X7 receptor constructs with 0 to 3 of the ATP binding sites. Each construct was then expressed in X. laevis oocytes and used to measure the relative current following ATP treatment. Response curves where then determined for each construct and demonstrated that having only one activation site is sufficient to elicit a response, while additional sites didn’t provide any further activation until all three where present. Overall the study is complete and thorough. However, the figures are extremely hard to read and can not be evaluated in their present form, making it difficult to adequately assess their findings. Several concerns should be addressed before being considered for publication.
Concerns:
- Figure 1, the labels for membrane residues are too small. Can not find the sites that are being discussed. Legend is also too small. Can’t read any of the words.
- Figure 2, It would be helpful to point out in the picture where each of the parts are. For example draw an arrow to show the S3 tag or label the extracellular space and intracellular space or the N-terminal versus the C-terminal
- Figure 3 and 4, text is too small
- Figure 5, 6, and 7, all of the lines are too small or too light that they don’t show up. Text is too small. Can’t read your findings therefore can’t evaluate your science.
Author Response
Figure 1, the labels for membrane residues are too small. Can not find the sites that are being discussed. Legend is also too small. Can’t read any of the words.
Thank you for your feedback on Figure 1. The original figure was created at a high resolution so that all details and residue labels were clearly visible when viewed at full size. Unfortunately, embedding the figure in the PDF for review may have reduced its resolution and made the labels less legible. The Protter tool used for this figure does not allow the font size of individual residue labels to be increased; enlarging the entire figure is the only way to increase label visibility. All key features, such as the Strep tags and the flexible GGS linkers, have been carefully annotated and color-highlighted. For the revised submission, we will provide Figure 1 as a separate high-resolution file and ensure that the legend and labels are clearly legible. Additionally, we have added a sentence (highlighted in yellow for revision) to the legend to explain the rational behind the additional numbering of the concatamers according to wild-type residue numbers.
Figure 2, It would be helpful to point out in the picture where each of the parts are. For example draw an arrow to show the S3 tag or label the extracellular space and intracellular space or the N-terminal versus the C-terminal.
Thank you for the suggestions. We have added the suggested labels. Additionally, we realized that we had inadvertently highlighted residues V66 as blue spheres instead of K64 residues in Fig. 2B, so we corrected this error. We also rotated the structure shown in Fig. 2B by 2 degrees vertically to align its perspective to that of the concatamer view in Fig. 2A.
Additionally, apart from the above comments, we made a minor revision of Fig. 2B. The gating residues S339 and S342 were supposed to be shown in cyan, but they were inadvertently displayed in yellow or gray due to the color scheme used for the transmembrane segments.
Figure 3 and 4, text is too small.
We have revised the figures accordingly and provided them as separate high resolution files.
Figure 5, 6, and 7, all of the lines are too small or too light that they don’t show up. Text is too small. Can’t read your findings therefore can’t evaluate your science.
We have revised the figures accordingly and provided them as separate high resolution files.
Reviewer 2 Report
Comments and Suggestions for Authors
The work by Markwardt et. al. extends our knowledge of activation in rP2X7 receptors. Whole- cell currents from concatenated receptors with disabled binding sites were systematically studied and complemented by biochemical and computational approaches. The work is very detailed performed and carefully presented. I have only some minor points.
2.2. Please provide the full linker sequence with the number of amino acids.
2.4. Can the authors give a number of how many oocytes were used for purification?
2.5. line 145 Why is the frequency for filtering higher than for recording?
Figure1
The quality of the figure is low. It is very difficult to read the legends in the figure. Please increase the size.
line 220 rat P2X7 has 595 amino acids. What is the difference in this study?
line 229 "The RMSD of 0.481 Å between the two structures indicates a high degree of structural
similarity"
Can the authors state which RMSD is critical and when the degree of structural similarity is low?
line 247 ...as stick and balls to their enhance their visibility...
Figure 4 A The numbering is misleading. Can you change the numbers so that K 64 gets 64 or specify the difference.
Figure 4 B - C is in the legend
line 361 Add the species for P2X4 for completeness.
line 407 and 410 "Stelmashenko, 2012 26031 /id" "Mol Pharmacol. 2012 410
Oct;82(4):760–766. doi: 10.1124/mol.112.080903"
update the citation format
Author Response
2.2. Please provide the full linker sequence with the number of amino acids.
Thank you for your suggestion. The complete sequences, including the P2X4- or P2X7-specific residues, are already shown color-coded and numbered in Figure 1A and B. The sequence of a single S3 tag (WSHPQFEK) is also provided in the legend to Figure 1 and described in detail with references on p. 2. For further enhance clarity, we have now included the complete GGS and Strep tag linker sequences in one-letter amino acid code, along with their respective lengths, in the Materials and Methods section of the revised manuscript.
2.4. Can the authors give a number of how many oocytes were used for purification?
Typically, we inject 12-15 Xenopus oocytes per experimental group. After incubation and labeling, up to 10 morphologically healthy oocytes per group are selected for purification using a stereomicroscope. The selected oocytes are homogenized together, and the homogenate is cleared by centrifugation. The cleared supernatant is then transferred to a fresh tube, carefully avoiding carryover of unsoluble material, and subjected to affinity chromatography. For SDS-PAGE analysis, the protein sample applied to each lane corresponds to an equivalent amount from the same number of oocytes, typically approximately one quarter to one half of an oocyte per lane.
We have added the corresponding information (highlighted in yellow) to the chapter "Biochemical visualization of the intactness of plasma membrane-bound P2X7 concatamers" on page 5.
2.5. line 145 Why is the frequency for filtering higher than for recording?
Thank you for the hint. You are right, normally, the filter frequency could be lower than the sampling rate. The 100 Hz filter frequency was the lowest available in our setup. Due to the slow current variability, we believe that the high filter frequency does not have a significant impact on our evaluation of the current kinetics.
Figure1
Question: The quality of the figure is low. Please increase the size.
Thank you for your feedback regarding the figure quality and legend legibilty. While the original figure was submitted at a relative high resolution, we have now doubled the image resolution for Figure 1. The updated figure should be easier to read.
line 220 rat P2X7 has 595 amino acids. What is the difference in this study?
Thank you for your helpful tip, we apologize for the typo, which we have corrected accordingly.
line 229 "The RMSD of 0.481 Å between the two structures indicates a high degree of structural similarity". Can the authors state which RMSD is critical and when the degree of structural similarity is low?
An RMSD (root-mean-square deviation) of less than 2.0 Å is widely accepted as indicative of high degree of structural similarity between protein structures, while values greater than 3.0 Å typically reflect significant structural differences. These thresholds are consistent with established criteria in structural biology, as supported by reference 46 (Peterson et al., 2009), which is cited in our manuscript. In our study, the observed overall RMSD of 0.481 Å between the concatamer and wild-type P2X7 structures is 4-6 times lower than these thresholds, robustly supporting our conclusion that the linkers and Strep tags do not cause any meaningful structural perturbations. Thus, the very low RMSD supports the conclusion that the overall structure of the concatamer remains virtually identical to the wild-type receptor.
line 247 ...as stick and balls to their enhance their visibility...
The sentence was changed to: “...as sticks to enhance their visibility...”
Figure 4 A The numbering is misleading. Can you change the numbers so that K 64 gets 64 or specify the difference.
Thank you for this comment. We have revised the legend of Fig. 4 (p. 13) and added the following sentence: “The arrow indicates the location of residue K64 in the wild-type rP2X7 monomers. Mutation of this residue to alanine results in the ATP-binding site knockout (7ko). Please note that the numbering in this panel corresponds to the modified construct that includes the linker and Strep tag insertions.
The revised text is highlighted in yellow.
Figure 4 B - C is in the legend
"C" has been changed to "B".
line 361 Add the species for P2X4 for completeness.
"P2X4" has been changed to "rP2X4".
line 407 and 410 "Stelmashenko, 2012 26031 /id" "Mol Pharmacol. 2012 410
Oct;82(4):760–766. doi: 10.1124/mol.112.080903".
update the citation format
The citation format has been changed accordingly.
Reviewer 3 Report
Comments and Suggestions for Authors
The authors use electrophysiology to address an unanswered question regarding occupancy of P2X ATP binding sites needed to cause channel opening and if there is cooperativity between these sites. Data are generally compelling and support that idea that binding at one site is sufficient for channel activation, but that additional binding alters channel kinetics. My specific comments/suggestion are detailed as follows:
Do the P2X7K64A concatamers alter channel facilitation? There is a long-standing, unexplained, observation that repeat short application of ~1mM ATP causes channel facilitation. The author’s concatamers may provide a means of addressing this channel property. Do single or double K64A concatamers display current facilitation with repeat brief ATP applications?
The naming, 7Ko (and use of “knockout” to describe a construct with a mutated ATP binding site), may be confusing to a broad audience. Since it is simply a point mutation, I’d suggest naming with the mutation. For example, a trimer concatamer with the second subunit ATP site disrupted would be X7-X7K64A-X7 and a concatamer with all three ATP sites disrupted would be X7K64A-X7K64A-X7K64A
Figure 1: Please provide the full vector sequence (concatamers as they are in the plasmid) for the concatamers in supplemental material.
Figure 3: To appropriately interpret this data, the following are needed: a total protein loading control is needed for quantification, a size marker is needed to confirm that bands are at approximately the expected size for one, two, or three subunit concatamers, and uninjected or mock injected (vector only or X7 lacking the Strep tag) is needed to assess specificity of strep tag. Clarification on why/how oocytes were determined to be “leaky” should also be provided.
Lines 267-282: Discussion/interpretation of the results here is distracting in my opinion. The important thing here is that expression of a dimer concatamer yields degradation fragments. You don’t see that with the trimer concatamer. Trimer bands are roughly equivalent between the two groups, thus it is unlikely that you’re missing a significant amount of degradation fragments in the trimer concatamer lanes (though still possible there is sum).
Figure 4: It would be more informative to show the residue in the full-length ATP bound rat P2X7 structure (PDB ID: 6U9W; https://doi.org/10.2210/pdb6U9W/pdb)
Figure 5: “Although the detailed effect of the knockouts on single-channel current and channel open probability is not known, the similar physical expression of all concatameric con- structs (Fig. 3) is reflected by comparable ATP-induced currents when the cRNA of the 7wt-7wt-7wt concatamer was diluted 1:10 for injection into X. laevis oocytes (Fig. 5D).”
This statement is unclear to me. It appears that all concatamers are functional, Conclusions about channel expression, Po, or conductance should be inferred or implied from this data.
How do whole cell currents from concatamers compare to injection of a single subunit construct? Presumably the single construct is much higher if the expression data is consistent with function.
Table 1 and Figure 6: Clarity could be improved by labeling the rates (A1, R1, A2, and R2) on an example trace in Figure 5.
Figure 7: It is unclear why a P2X4 trimeric concatamer was also used. Please provide rationale.
Minor:
Line 132: Please correct “Coomassie-fre” to “Coomassie-free”
Lines 403-411: Please correct issues with references.
Author Response
Do the P2X7K64A concatamers alter channel facilitation? There is a long-standing, unexplained, observation that repeat short application of ~1mM ATP causes channel facilitation. The author’s concatamers may provide a means of addressing this channel property. Do single or double K64A concatamers display current facilitation with repeat brief ATP applications?
Thank you for raising this important point. Our experiments were not specifically designed to address this issue. In our protocol, we applied different concentrations of ATP every 2 min for 6 sec, with each ATP application preceded by a 0.1 mM ATP application. When we then compared the current amplitudes of the second and third 0.1 mM ATP applications with those of the first application, we found no statistical difference between the concatameric constructs and the non-concatameric rP2X7 constructs.
The naming, 7Ko (and use of “knockout” to describe a construct with a mutated ATP binding site), may be confusing to a broad audience. Since it is simply a point mutation, I’d suggest naming with the mutation. For example, a trimer concatamer with the second subunit ATP site disrupted would be X7-X7K64A-X7 and a concatamer with all three ATP sites disrupted would be X7K64A-X7K64A-X7K64A
Thank you for your suggestion regarding the naming convention. We have chosen to use "KO" in our construct names for several reasons: (i) our primary focus is on the functional knockout of ATP binding, rather than the precise amino acid position mutated - especially since there are multiple possible sites for such a knockout; (ii) the abbreviation "KO" is widely and intuitively understood by most readers; and (iii) "KO" is concise and easier to use in figures and text than, for example, "K64A". Importantly, we have clearly stated in the manuscript that the K64A mutation was introduced, so there should be no ambiguity for the reader. In addition, in Figure 1, which shows the transmembrane topology of the concatamers, we have labeled the color-highlighted position of K64 (wt numbering). Last but not least, we tried to keep the figure labels short, otherwise we would have to use smaller characters, which might be less readable.
Figure 1: Please provide the full vector sequence (concatamers as they are in the plasmid) for the concatamers in supplemental material.
As suggested, we provide the complete vector sequences of the concatamers shown in Fig. 3 in Supplementary Table 2. To assign each vector sequence to the expressed protein, we use our internal laboratory construct numbers, which are now also shown in the legend of Fig. 3 (bottom row).
Figure 3: To appropriately interpret this data, the following are needed: a total protein loading control is needed for quantification, a size marker is needed to confirm that bands are at approximately the expected size for one, two, or three subunit concatamers, and uninjected or mock injected (vector only or X7 lacking the Strep tag) is needed to assess specificity of strep tag. Clarification on why/how oocytes were determined to be “leaky” should also be provided.
Thank you for your detailed suggestions. We address each point below:
- Total protein loading control: The protein loaded in these lanes was selected for analytical purposes. All samples were highly purified by StrepTactin chromatography, resulting in gel scans where only the expressed exogenous proteins (confirmed by their expected masses) are visible, not comparable to Western blots. Exceptions to the expected masses occurred only when rP2X7 homodimers were expressed, resulting in partial generation of monomeric by-products, apparently by proteolytic cleavage, and closely related to our previous experience with P2X1 concatamers (Nicke et al., Mol Pharmacol 2003, cited in Discussion). Loading controls are not necessary here, as this experiment was designed to verify plasma membrane expression of the candidate protein at its expected mass, and in particular the unwanted presence of lower order by-products (e.g., monomers/dimers in trimer concatamers) that could confound electrophysiological interpretations, as we have previously shown (Nicke et al., Mol Pharmacol 2003). For concatenated rP2X7 homotrimers, all lanes show only the expected proteins as indicated by their expected masses, confirming the absence of significant by-products. Concatamers are widely used in electrophysiological experiments, and are increasingly being tested for their intactness, as they should be.
- Mass markers: We did not include commercial mass markers in this particular gel because we are very familiar with the migration behavior of these proteins from extensive previous experiments. Our primary goal was to compare as many rP2X7 mutants as possible on a single gel for publication, limited only by the number of lanes. In this context, we consider the “non-concatenated” rP2X7 samples in lanes 1-2 to be ideal internal mass references. These samples are the direct precursors of the concatamers (differing only in linker sequences and concatenation) and consistently show the characteristic band pattern for monomer, homodimer, and homotrimer forms of a partially denatured rP2X7 receptor, as previously published (e.g., Becker et al., 2008). The relative migration of these non-concatenated rP2X7 bands is well known by and typical for these constructs and closely reflects the behavior of the concatamers. For clarity, we have now added the calculated masses corresponding to the non-concatenated rP2X7 bands directly to the figure.
- Specificity of the Strep tag: The use of StrepTactin purification in combination with the Strep Tactin double tag ensures that only proteins containing the Strep tag are isolated. We have extensive experience with this system and are not aware of any non-specific binding of endogenous oocyte proteins under the conditions used. The clear banding pattern, the absence of additional bands, and the use of mock-injected or non-tagged controls in related experiments confirm the specificity of the detection. Due to the highly selective nature of the StrepTactin system and the analytical purpose of this experiment, additional mock-injected controls were not included.
- Definition of "Leaky" Oocytes: Oocytes were classified as "leaky" based on (1) the experimenter's observation during the experiment that one or more oocytes were damaged during incubation with covalently reacting IR 800 dye, resulting in cytoplasmic leakage, and (2) the appearance of high molecular mass bands that, likely represent aggregates of the expressed proteins. Due to the time-sensitive nature of the experiments, with many groups of oocytes processed in a single day, including PAGE analysis, it is not feasible to check the integrity of each oocyte individually. To avoid drawing erroneous conclusions from data potentially affected by leaky oocytes, we excluded such lanes from the final analysis. However, we did not cut them out, preferring to preserve the integrity of the gel and avoid any discussion of gel manipulation.
Summary: The combination of highly specific purification, clear and expected banding patterns, and extensive prior experience with these constructs provides robust evidence for the identity and purity of the expressed proteins. While we appreciate the importance of controls in quantitative Western blotting, in this analytical context, where we are analyzing highly purified proteins in a system that we know very well, the additional controls requested would not add further value. We hope this clarifies our rationale and the robustness of our approach. In addition, the expressed non-concatenated rP2X7 receptor monomers provide a clear band pattern of ideal mass markers, allowing us to assess the masses of the concatamers and byproducts.
Lines 267-282: Discussion/interpretation of the results here is distracting in my opinion. The important thing here is that expression of a dimer concatamer yields degradation fragments. You don’t see that with the trimer concatamer. Trimer bands are roughly equivalent between the two groups, thus it is unlikely that you’re missing a significant amount of degradation fragments in the trimer concatamer lanes (though still possible there is sum).
Thank you for your question. The rat P2X7 receptor indeed comprises 595 amino acids (UniProt Q64663). We confirm that our concatameric constructs include the full-length rat P2X7 coding sequence (all 595 amino acids) for each subunit, with additional flanking linker and Strep tag sequences. No truncation or deletion of the P2X7 sequence occurred in these constructs; the linker and tag sequences were added solely to promote concatamer folding and purification, respectively.
As discussed in the manuscript (line 286 ff.), the observed monomers likely arise from frustrated assembly into a pseudo-tetramer composed of two dimeric rP2X7 concatamers. This incomplete assembly triggers ER-associated degradation of unincorporated subunits - similar to a “leave-one-out” mechanism. This interpretation is consistent with our previous findings for P2X1 concatamers (Nicke et al., 2003), which are cited and contextualized in the manuscript.
Figure 4: It would be more informative to show the residue in the full-length ATP bound rat P2X7 structure (PDB ID: 6U9W; https://doi.org/10.2210/pdb6U9W/pdb)
Thank you for this suggestion. The “K64” residue is now marked in Fig. 2. The residue within the ATP binding pocket is also shown in Oken A.C.; Lisi N.E.; Krishnamurthy I.; McCarthy A.E.; Godsey M.H.; Glasfeld A.; Mansoor S.E. High-affinity agonism at the P2X7 receptor is mediated by three residues outside the orthosteric pocket. Nat. Commun. 2024, 15, 6662; https://doi.org/10.1038/s41467-024-50771-6.
Figure 5: “Although the detailed effect of the knockouts on single-channel current and channel open probability is not known, the similar physical expression of all concatameric constructs (Fig. 3) is reflected by comparable ATP-induced currents when the cRNA of the 7wt-7wt-7wt concatamer was diluted 1:10 for injection into X. laevis oocytes (Fig. 5D).”
This statement is unclear to me. It appears that all concatamers are functional, Conclusions about channel expression, Po, or conductance should be inferred or implied from this data.
Thank you for your feedback. We agree that whole-cell currents (I=N×Po×i) reflect the combined effects of channel number (N), open probability (Po), and single-channel current (i). Figure 3 shows comparable expression levels (N), and Figure 5D shows similar current amplitudes (II) across constructs. This implies that the product Po × i is not significantly altered by the knockout mutations. However, as you rightly point out, we cannot determine the individual contributions of Po and i without single-channel recordings. We have revised the text to remove speculative claims about Po or i and to focus on the functional comparability of the constructs.
The corresponding paragraph was modified accordingly.
How do whole cell currents from concatamers compare to injection of a single subunit construct? Presumably the single construct is much higher if the expression data is consistent with function.
The maximum current amplitude, Imax, when single rP2X7 subunits are expressed by injection of a 1:500 diluted cRNA, is approximately equal to the Imax values of the knockout constructs. This suggests a greatly reduced functional expression due to concatamerization, corresponding to the greatly reduced physical expression of rP2X7 concatamers compared to rP2X7 monomers.
Table 1 and Figure 6: Clarity could be improved by labeling the rates (A1, R1, A2, and R2) on an example trace in Figure 5.
We have revised the figure accordingly and added the following sentence to the legend: "In A, the characters a, b, c and d mark the time course of fast and slow activation and fast and slow deactivation, respectively."
Figure 7: It is unclear why a P2X4 trimeric concatamer was also used. Please provide rationale.
The rP2X4 concatamer data show that the functionally distinct activation sites are not a specific property of P2X7 receptors. A corresponding remark was inserted in the discussion.
Minor:
Line 132: Please correct “Coomassie-fre” to “Coomassie-free”
Thank you for pointing out the typo. “Coomassie-fre” has been corrected to “Coomassie-free.”
Lines 403-411: Please correct issues with references.
Thank you for pointing out the problems with the references. These have been fixed as requested.
Reviewer 4 Report
Comments and Suggestions for Authors
Using concatamers of wild-type and ATP-binding site knockout (K64A mutant) rat P2X7 receptors (P2X7R), this study reveals novel insights to the opening and kinetics of rat P2X7R upon binding of the trimeric receptor to 1, 2 or 3 ATP molecules. The study appears to be well executed and is clearly presented. However, a number of relatively minor points need to be addressed before this manuscript can be considered for publication.
- Title: Given the differences between rat and human P2X7R discussed within the manuscript, it would be preferable if the title included "rat".
- Abstract: To help improve visibility, the authors may wish to include a comment about the rat P2X4 receptor data presented within the manuscript. Or at minimum add "P2X4 receptor" to the keywords.
- Abstract, line 13: italicize Xenopus oocytes.
- Lines 31-32: Correct "Danger-Associated Molecular Patterns (DAMPs)" to "danger-associated molecular pattern (DAMP)"
- Lines 36-37: Please give more thought to the inconsistent use of "Ca2+- and Mg2+-free media [8,9]. In contrast, in divalent-free solutions..." It may be clearer/simpler if "divalent-free solutions" were used in both sentences.
- Line 50: "rP2X7R" should be introduced here not at line 80. Revise both lines accordingly.
- Line 56: The sentence "Although apparently structurally identical..." Identical to what? rP2X7R?
- Line 69: Correct "X. laevis" to "Xenopus laevis".
- Line 149: Given comment on line 36 about Mg2+-free media and given this cation is often present in P2X7R solutions, it seems insufficient not to list Mg2+ here along with Ca2+. Consider revising.
- rP2X1 (line 195) and rP2X4 (line 201) should be defined in full, as for rP2X7. Check manuscript for any other similar inconsistencies.
- Line 200: Delete "(wt") as "wild-type" used subsequently (except lines 433 and 438, which can be changed to "wild-type").
- Line 260: "expression level" should be reserved for mRNA not proteins. Revise accordingly.
- Line 410-411: Insert citation correctly.
- Lines 415 and 423: Correct "P2X7" to "P2X7R" and "hP2X7 receptor" to "hP2X7R", respectively. Check manuscript for any other similar inconsistencies.
- Line 424-428: Given the conclusions regarding rP2X7R (e.g. lines 20-21), this discussion seems to be about hP2X7 but this is unclear. Revise accordingly.
- Line 464: Correct "human P2X7 receptors" to "hP2X7R".
- The supplementary files does not contain Supplementary Fig. 3 (line 266) nor a Supplementary Fig. 2 for that matter (not cited). Correct accordingly and ensure all figures and their panels are cited.
- Whilst I appreciate Cells is an online only journal, the figures, especially Figures 1, 4-7 are quite difficult to read when printed. The authors may wish to improve the size of some fonts as required.
Author Response
- Title: Given the differences between rat and human P2X7R discussed within the manuscript, it would be preferable if the title included "rat".
As suggested, the title was changed to "Activation of the rat P2X7 receptor by functionally different ATP activation sites" to clearly indicate the focus on the rat P2X7 receptor and to reflect the species-specific differences discussed in the manuscript.
- Abstract: To help improve visibility, the authors may wish to include a comment about the rat P2X4 receptor data presented within the manuscript. Or at minimum add "P2X4 receptor" to the keywords.
As suggested, the keyword "P2X4 receptor" has been added to the keyword list.
- Abstract, line 13: italicize Xenopus oocytes.
As suggested, we have italicized “Xenopus”.
- Lines 31-32: Correct "Danger-Associated Molecular Patterns (DAMPs)" to "danger-associated molecular pattern (DAMP)"
We have corrected this term as suggested.
- Lines 36-37: Please give more thought to the inconsistent use of "Ca2+- and Mg2+-free media [8,9]. In contrast, in divalent-free solutions..." It may be clearer/simpler if "divalent-free solutions" were used in both sentences.
Thank you for your suggestion. We have changed the sentence to "In contrast, in such divalent-free solutions, …"
- Line 50: "rP2X7R" should be introduced here not at line 80. Revise both lines accordingly.
We have fixed this as suggested.
- Line 56: The sentence "Although apparently structurally identical..." Identical to what? rP2X7R?
We have the sentence to read: "Although the three ATP binding sites of the human P2X7R (hP2X7R) are apparently structurally identical, we have previously observed that they contribute differentially to receptor activation…"
- Line 69: Correct " laevis" to "Xenopus laevis".
We have fixed this as suggested.
- Line 149: Given comment on line 36 about Mg2+-free media and given this cation is often present in P2X7R solutions, it seems insufficient not to list Mg2+ here along with Ca2+. Consider revising.
We changed "This Ca2+-free solution …" to "This Ca2+- and Mg2+-free solution".
- rP2X1 (line 195) and rP2X4 (line 201) should be defined in full, as for rP2X7. Check manuscript for any other similar inconsistencies.
We have added the species name accordingly.
- Line 200: Delete "(wt") as "wild-type" used subsequently (except lines 433 and 438, which can be changed to "wild-type").
Fixed as suggested.
- Line 260: "expression level" should be reserved for mRNA not proteins. Revise accordingly.
Thank you for your comment regarding the use of the term "expression level". While we appreciate your perspective, we would like to point out that "expression level" is widely used in the literature to describe both mRNA and protein abundance. For example, numerous studies and reference databases refer to "protein expression levels" when discussing results from Western blotting, immunohistochemistry, or other protein quantification methods. We therefore believe that our use of the term in reference to protein levels is consistent with standard scientific practice. However, if the editors prefer, we are happy to clarify the text by specifying "protein expression level" where appropriate.
- Line 410-411: Insert citation correctly.
The citation has been inserted in the correct location as requested.
- Lines 415 and 423: Correct "P2X7" to "P2X7R" and "hP2X7 receptor" to "hP2X7R", respectively. Check manuscript for any other similar inconsistencies.
We have corrected "P2X7" to "P2X7R" and "hP2X7 receptor" to "hP2X7R" in the indicated lines and have reviewed the entire manuscript for consistency in nomenclature.
- Line 424-428: Given the conclusions regarding rP2X7R (e.g. lines 20-21), this discussion seems to be about hP2X7 but this is unclear. Revise accordingly.
Thank you for pointing out the ambiguity. Our hypothesis refers to P2X7 receptors in general, not specifically to the human or rat isoform. We have revised the relevant sentences to clarify this and to ensure that the discussion consistently refers to P2X7 receptors as a receptor family where appropriate.
- Line 464: Correct "human P2X7 receptors" to "hP2X7R".
Thank you for your comment. "Human P2X7 receptors" has been changed to "hP2X7Rs" as requested.
- The supplementary files does not contain Supplementary Fig. 3 (line 266) nor a Supplementary Fig. 2 for that matter (not cited). Correct accordingly and ensure all figures and their panels are cited.
We apologize for our mistake and thank you for pointing it out. We have fixed it accordingly.
- Whilst I appreciate Cellsis an online only journal, the figures, especially Figures 1, 4-7 are quite difficult to read when printed. The authors may wish to improve the size of some fonts as required.
We have revised the figures accordingly and provide them as separate high resolution files.
Round 2
Reviewer 1 Report
Comments and Suggestions for Authors
Again the text throughout most of the figures are still too small, such that they cannot be read. Figure 1 is impossible to read. Using subscripts is unacceptable (Fig ). Figure 6C is cut off and incomplete.
Author Response
Comments and Suggestions for Authors
Again the text throughout most of the figures are still too small, such that they cannot be read. Figure 1 is impossible to read. Using subscripts is unacceptable (Fig ). Figure 6C is cut off and incomplete.
We apologize for this inconvenience. Originally we supplied images in JPEG format with 600 dpi resolution. Unfortunately, the resolution drops after incorporation of these figure files into the Word manuscript. We have now generated and incorporated TIFF files.
The cut-off in Figure 6 has now been repaired.
Reviewer 3 Report
Comments and Suggestions for Authors
"We have chosen to use "KO" in our construct names for several reasons: (i) our primary focus is on the functional knockout of ATP binding, rather than the precise amino acid position mutated - especially since there are multiple possible sites for such a knockout"
This is why you should clarify the site and mutation rather than obscure it with the phrase "knockout" - especially in the abstract.
Author Response
"We have chosen to use "KO" in our construct names for several reasons: (i) our primary focus is on the functional knockout of ATP binding, rather than the precise amino acid position mutated - especially since there are multiple possible sites for such a knockout"
This is why you should clarify the site and mutation rather than obscure it with the phrase "knockout" - especially in the abstract.
The abstract now explicitly states that up to three ATP binding sites were knocked out by the K64A mutation.